# Development of a Potential Gallium-68-Labelled Radiotracer Based on DOTA-Curcumin for Colon-Rectal Carcinoma: From Synthesis to In Vivo Studies

**DOI:** 10.3390/molecules24030644

**Published:** 2019-02-12

**Authors:** Giulia Orteca, Federica Pisaneschi, Sara Rubagotti, Tracy W. Liu, Giacomo Biagiotti, David Piwnica-Worms, Michele Iori, Pier Cesare Capponi, Erika Ferrari, Mattia Asti

**Affiliations:** 1Department of Chemical and Geological Sciences, University of Modena and Reggio Emilia, via G. Campi 103, 41125 Modena, Italy; giulia.orteca@unimore.it; 2Department of Cancer Systems Imaging, The University of Texas MD Anderson Cancer Center, 1881 East Road, Houston, TX 77054, USA; FPisaneschi@mdanderson.org (F.P.); twliu@mdanderson.org (T.W.L.); biagio2588@gmail.com (G.B.); dpiwnica-worms@mdanderson.org (D.P.-W.); 3Radiopharmaceutical Chemistry Unit, Oncologic and High Technologies Department, Azienda USL-IRCCS, via Amendola 2, 42122 Reggio Emilia, Italy; sara.rubagotti@libero.it (S.R.); michele.iori@ausl.re.it (M.I.); piercesare.capponi@ausl.re.it (P.C.C.); mattia.asti@ausl.re.it (M.A.); 4Department of Chemistry “Ugo Schiff”, Università di Firenze, via della Lastruccia 3-13, 50019 Sesto Fiorentino, Italy

**Keywords:** gallium-68, radiotracers, positron emission tomography, curcuminoids, colorectal cancer

## Abstract

Colorectal cancer is the third most commonly occurring cancer in men and the second most commonly occurring cancer in women worldwide. We have recently reported that curcuminoid complexes labelled with gallium-68 have demonstrated preferential uptake in HT29 colorectal cancer and K562 lymphoma cell lines compared to normal human lymphocytes. In the present study, we report a new gallium-68-labelled curcumin derivative (^68^Ga-DOTA-C21) and its initial validation as marker for early detection of colorectal cancer. The precursor and non-radioactive complexes were synthesized and deeply characterized by analytical methods then the curcuminoid was radiolabelled with gallium-68. The in vitro stability, cell uptake, internalization and efflux properties of the probe were studied in HT29 cells, and the in vivo targeting ability and biodistribution were investigated in mice bearing HT29 subcutaneous tumour model. ^68^Ga-DOTA-C21 exhibits decent stability (57 ± 3% after 120 min of incubation) in physiological media and a curcumin-mediated cellular accumulation in colorectal cancer cell line (121 ± 4 KBq of radiotracer per mg of protein within 60 min of incubation). In HT29 tumour-bearing mice, the tumour uptake of ^68^Ga-DOTA-C21 is 3.57 ± 0.3% of the injected dose per gram of tissue after 90 min post injection with a tumour to muscle ratio of 2.2 ± 0.2. High amount of activity (12.73 ± 1.9% ID/g) is recorded in blood and significant uptake of the radiotracer occurs in the intestine (13.56 ± 3.3% ID/g), lungs (8.42 ± 0.8% ID/g), liver (5.81 ± 0.5% ID/g) and heart (4.70 ± 0.4% ID/g). Further studies are needed to understand the mechanism of accumulation and clearance; however, ^68^Ga-DOTA-C21 provides a productive base-structure to develop further radiotracers for imaging of colorectal cancer.

## 1. Introduction

Colorectal cancer (CRC) is the third most commonly occurring cancer in men and the second in women worldwide, with more than 1.4 million new cancer cases every year [1]. India has a relatively low prevalence of CRC in comparison to other countries, mainly attributed to differences in dietary patterns and lifestyles [2]. It is widely known that curcumin (1,7-bis(4-hydroxy-3-methoxyphenyl)-1,6-heptadiene-3,5-dione), a phyto-polyphenol pigment isolated from the dried rhizomes of *Curcuma longa* L., has been used for centuries in Asian countries as a dietary spice and traditional drug to treat many different diseases [3]. Besides having antioxidant, anti-inflammatory, anti-bacterial and anti-amyloid properties, curcumin has been shown also to possess anti-cancer activities [4,5]. In particular, the ability to inhibit proliferation of a wide variety of tumour types, including colorectal cancer, has been highlighted by recent studies in vitro and in vivo [6,7,8]. Several studies suggest that curcumin changes the gene expression profiles and signalling pathways such as those involving COX-2 enzymes as well as NF-κB, cyclin D1 and p53 proteins [9,10,11]. Although the mechanism of curcumin’s high accumulation and cytotoxic activity against HeLa, MCF-7, HT29, and HCT-116 cancer cell lines are still unclear [10,11], curcumin itself is a promising tumour-targeting moiety for the development of early diagnosis and therapeutic agents. Nuclear medicine imaging provides information about biological processes occurring at a molecular level in vivo; this occurs by following the fate of radiolabelled compounds. Such radiotracers can be substrates for metabolic pathways overexploited in tumour cells or have particular affinity for receptors overexpressed in tumours. Hence, curcumin derivatives labelled with proper β^+^ emitter radionuclides or curcumin-based radio-metal complexes have the potential to be diagnostic tools for positron emission tomography (PET). Currently, radiolabelled curcuminoids have been mostly investigated for imaging of the central nervous system (CNS) where several curcumin-based probes labelled with fluorine-18 have demonstrated promise in the early detection of Alzheimer’s disease [12,13]. In oncology, curcuminoids as imaging agents are underinvestigated. We have recently reported that curcuminoid complexes labelled with gallium-68 demonstrated preferential uptake in HT29 colorectal cancer and K562 lymphoma cell lines compare to normal human lymphocytes. In that study, the core radio-metal was directly linked to two curcuminoid molecules through the keto-enol moiety [14,15]. The curcuminoid complexes showed good stability in PBS and human serum but were rapidly degraded in whole human blood. These findings, in addition to the low stability and solubility of curcumin structures in physiological media [16], limited the possibility to further study these kind of compounds in vivo. In the present study, we report a new gallium-68-labelled curcumin derivative with curcumin linked to an efficient gallium-68 chelator, 1,4,7,10-tetraazacyclododecane-1,4,7,10-tetraacetic acid (DOTA), and its initial validation as a marker for early detection of CRC. The synthetic pathway, chemical characterization, labelling methods, stability, cell uptake, internalization and efflux studies on HT29 cells and in vivo micro-PET imaging and biodistribution in colorectal tumour bearing mice of this radiotracer are herein reported.

## 2. Results

### 2.1. Synthesis and Chemical Characterization

Synthesis of 1,4,7,10-tetraazacyclododecane-1,4,7,10-tetraacetic(1,7-bis(4-hydroxy-3-methoxyphenyl)-1,6-heptadiene-3,5-dione (DOTA-C21) was carried out in two steps as shown in Figure 1. The curcumin-based conjugate was identified and completely characterized by ESI-LC/MS (Appendix A) and ^1^H-,^13^C-NMR spectroscopy (Appendix A). ESI-LC/MS spectrum exhibits two isomeric peaks (*m/z* = 798.3; [M + H]^+^) that were attributed to contemporary presence of two main isomeric forms (diketo and keto-enol) as previously reported for curcumin and its derivatives [17,18]. Gallium complexes were obtained by reacting a DOTA-C21 solution with a Ga(NO_3_)_3_ solution at 95 °C for 30 min in a 1.5:1 metal to ligand molar ratio using a 0.4 M ammonium acetate solution as buffer. Complexes formation was identified by ESI-LC/MS analysis thanks to the typical isotopic pattern of gallium ion (Appendix A). Similar to the free ligand analysis, the spectrum exhibits two compounds with the same *m/z* (retention times: 2.8 and 3.3 min, respectively) both corresponding to Ga-DOTA-C21 complexes (*m/z* = 864.2 [M^69^Ga + H]^+^ – 866.2 [M^71^Ga + H]^+^). To elucidate chelation mode and binding sites, ^1^H-NMR titration of DOTA-C21 with Ga^3 +^ was also performed (Appendix A) and the gallium(III)-complex was characterized by NMR (Figure 2, Appendix A) as well as LC-MS/MS fragmentation experiments (Figure 3, Appendix A).

^1^H- and ^13^C-NMR spectra displayed a strong shift of both protons and carbon with respect to the free ligand when gallium(III) is added. In particular, methylene protons of the chelator ring, briefly *r_1_* and *r_2_*, are no longer chemical shift equivalent since the interconversion between pseudo-axial (H_a_) and pseudo-equatorial (H_e_) conformation is prevented by metal complexation. Geminal protons H_a_(*r_1_*) and H_e_(*r_1_*) appear as broad triplets at 4.00 ppm and 3.38, respectively, due to similar values of geminal (^2^J_HH_) and vicinal (^3^J_HH_) coupling constants (~14Hz). The same situation occurs for H_a_(*r_2_*) and H_e_(*r_2_*) that fall at 3.47 ppm and 3.41 ppm, respectively. Assignments of ring protons was confirmed by ^1^H,^13^C-HSQC spectrum, H_a_/H_e_(*r_1_*) show cross-peaks with carbon at 54.65 ppm (C(*r_1_*)) and H_a_/H_e_(*r_2_*) with a carbon at 57.11 ppm (C(*r_2_*)). The methylene groups of acetate arms are not chemical shift equivalent in gallium complex and appear as broad singlets in proton spectrum. The following reasonable assignments can be proposed (for atom numbering refers to Section 4.1.1 and Section 4.1.2): CH_2_ in α to amide and nitrogen (atom numbering 15) ^1^H- 3.79 ppm and ^13^C- 60.64 ppm; CH_2_ in α to carboxylic group involved in gallium coordination (atom numbering 24 and 26) ^1^H- 3.83 ppm and ^13^C- 59.41 ppm; CH_2_ in α to free carboxylic group (atom numbering 25) ^1^H- 3.74 ppm, ^13^C- 61.68 ppm.

Moreover, UV absorption and fluorescent emission spectra upon increasing addition of Ga^3+^ from free ligand were acquired (Appendix A). These analyses showed that DOTA-C21 displays an absorption maximum at 410 nm, typical of curcumin structure in the keto-enol form. Upon addition of gallium (III) solution, an increase in absorbance is observed at 290 nm, while no changes are observed for the absorption maximum. Fluorescence emission spectra of DOTA-C21 (λ_ex_ 410 nm) do not show any significant change after gallium(III) addition.

### 2.2. Radiolabelling of the Chelator-Curcuminoid Derivatives with Gallium-68

Radiolabelling was performed using two methods. In the first, postprocessing of the generator elution was performed by cation exchange purification [19] and kinetics of the labelling were studied between 1 and 20 min by TLC. Kinetics were fast and the incorporation yield was 97 ± 2% after 5 min at 95 °C (*n* = 3). Postprocessing of the generator elution minimizes the metal cation impurities allowing a complete radiolabelling of DOTA-C21 by using only 10 nmol of the precursor. Representative TLC analyses are shown in Appendix A. In the second method, labelling without processing of the elution was attempted with increasing amount of precursor (20, 40, 60, 80 nmol). In this case, a reproducible radiolabelling could only be achieved with 80 nmol of DOTA-C21 after 10 min at 95 °C, with an incorporation yield ranging from 60 to 80%. The crude mixture was further purified by SPE to eliminate unreacted gallium-68 and hydrolysed products. Radiochemical purity was assessed by UHPLC (RCP > 95%, RCY = 55 ± 5%, *n* = 10). Representative UHPLC chromatograms (radiochemical detector) of a ^68^Ga-DOTA-C21 preparation before and after SPE purification are shown in Figure 4. Similarly, to the complexation reaction with natural gallium, two main radioactive species were obtained with a retention time of 4.7 and 5.1 min, respectively. Based on the data gathered for the non-radioactive preparation, the peaks were assigned to two ^68^Ga-DOTA-C21 isomeric complexes. A further peak at 5.3 min was recorded (ca. 4% of total radioactivity) and was likely due to a gallium-68 complexes with a degraded form of the precursor at labelling conditions. In both methods, the pH was kept constant around 4.

### 2.3. In Vitro Stability Studies

In vitro stability of ^68^Ga-DOTA-C21 complexes in PBS, human serum (HS) and whole human blood (HB) were determined according to peak integration of analytical UHPLC (Figure 5). Stability of ^68^Ga-DOTA-C21 in PBS was more than 88 ± 3% after 120 min at 37 °C. In HS, the radiotracer was also stable with 85 ± 3% of intact compound after 120 min of incubation. On the contrary, in whole blood, ^68^Ga-DOTA-C21 was partially degraded to more polar metabolites already after 10 min of incubation. The amount of intact compound ranged from 74 ± 2% after 10 min to 57 ± 3% after 120 min of incubation.

### 2.4. Uptake, Internalization and Efflux in Colorectal Cancer Cell Line

In vitro ^68^Ga-DOTA-C21 uptake studies demonstrated time-dependent cellular accumulation in HT29 colorectal cancer cell line. Total uptake increased continuously in the first 60 min of incubation up to 121 ± 4 KBq of radiotracer per mg of protein (*n* = 3). At 60 min post incubation, 83% of total radioactivity was internalized in the cells. Conversely, ^68^GaCl_3_, used as a negative control, demonstrated significantly lower in vitro cell uptake (20 ± 4 KBq/mg of protein after 60 min of incubation, *n* = 3) confirming the curcuminoid-mediated uptake by HT29 cells (Figure 6A,B). The cellular uptake of ^68^Ga-DOTA-C21 is comparable to that of the compound obtained by direct labelling of curcumin with gallium-68 (i.e., ^68^Ga(CUR)_2_ complexes [15]), however the internalized fraction was more than doubled. To test for receptor-mediated uptake, HT29 cells were preincubated with a 200-fold excess of calcitriol, the natural ligand for vitamin D receptor (VDR), and then ^68^Ga-DOTA-C21 was added. The experiments were performed to assess whether ^68^Ga-DOTA-C21 uptake was mediated by VDR since it has been reported that curcumin has a structure suitable to bind nuclear vitamin D receptor. The difference between uptake with or without calcitriol was not statistically significant (*n* = 3, Figure 6C) suggesting that the VDR receptor was not involved in the uptake or that the receptor cannot be saturated in these conditions.

For efflux studies, cells were incubated with ^68^Ga-DOTA-C21 and the uptake was stopped after 60 min by removing the supernatant. Then, cells were washed, fresh medium was added and they were incubated again for different periods of time. Media were collected and radioactivity released from the cells was measured.^68^Ga-DOTA-C21 showed a slow externalization pattern with around 75% (*n* = 3) of the radiotracer remaining within the cell after 60 min of incubation (Figure 7).

### 2.5. PET Imaging and Biodistribution

HT29 tumour-bearing mice were injected i.p. with ^68^Ga-DOTA-C21 and PET/CT imaging was taken 1 h and 2 h post injection (Figure 8). The following day, the same cohort of mice was injected i.v. with ^68^Ga-DOTA-C21 and dynamic PET/CT imaging was registered for the first 10 min to assess initial distribution of the tracer. The 1 h post injection time point was retaken to evaluate potential differences in the biodistribution pathway related to the type of injection. Mice were sacrificed at 1.5 h post injection, and ex-vivo biodistribution performed. As per PET/CT imaging analysis, upon i.v. injection, ^68^Ga-DOTA-C21 was taken up rapidly in the heart and liver, and began accumulating into the tumour within the first minute of circulation. The tumour was immediately visible over background (muscle). At 1 h post injection, tumour accumulation of the tracer was 2.27 ± 0.85% ID/cc (*n* = 3) with a tumour to muscle ratio (T/M) of 1.91 ± 0.47 (*n* = 3). ^68^Ga-DOTA-C21 appeared to have both renal and hepatic clearance. Radioactivity in the heart, which reflected the blood pool, was surprisingly high at 1 h post injection (5.8 ± 0.7% ID/cc, *n* = 3). Similar biodistribution was observed when the tracer was delivered i.p. At 1 h post injection, the tumour had an uptake of 1.89 ± 0.45% ID/cc (*n* = 5) and a T/M of 3.59 ± 2.76 (*n* = 5). At 2 h post injection, tumour uptake increased to 2.60 ± 0.27% ID/cc (*n* = 5) which was reflected by an increase in the T/M to 5.41 ± 3.62 (*n* = 5) (Figure 9 and Appendix A). It is worth noting that, when analysing tumour uptake in each single animal, there was no appreciable variation between 1 h and 2 h post injection in animals with an initial low tumour uptake. On the contrary, animals with a tumour uptake greater than 4% ID/cc after one hour, tended to accumulate more radiotracer (Appendix A). No correlation was found between T/M ratio and tumour size (Appendix A). Ex-vivo biodistribution confirmed the described accumulation (Figure 9). Tumour uptake was 3.08 ± 0.53% ID/cc (*n* = 5) with a T/M of 2.29 ± 0.62 (*n* = 5). As already observed by PET, clearance occurred via both renal and hepatobiliary excretion, bone uptake was barely detectable over background and no significant radioactivity was seen in the intracranial region, suggesting that ^68^Ga-DOTA-C21 is unlikely to cross an intact blood–brain barrier. Blood radioactivity was high, suggesting that a high amount of the radiotracer was in circulation or bound to blood constituents such as serum albumin. Further studies evaluating the clearance mechanism of ^68^Ga-DOTA-C21 are underway to understand the high blood circulation, however one explanation may be that ^68^Ga-DOTA-C21 pharmacokinetics follow a two-compartment model [20,21].

### 2.6. Metabolism

Mouse blood was drawn 1.25–1.5 h post injection, processed and analysed by radio-HPLC. Out of the two isomers present in the parent compound, one disappears in the blood and only the more polar one (Rt = 7.7 min) persists in all the analysed mice (*n* = 5) (Appendix A). It is likely that physiological media influence the keto-enol tautomerism of ^68^Ga-DOTA-C21, stabilizing the diketo form [18].

## 3. Discussion

The use of curcumin and its derivatives as potential therapeutics in many medical applications has been largely attested by a long list of scientific literature [22,23,24]. We previously demonstrated that ^68^Ga-labelled curcumin complexes had preferential accumulation in HT29 colorectal cancer cells compared to other cancer cell lines and lymphocyte cultures [15]. However, the stability of these complexes in blood was low as the curcuminoid backbone was directly linked to the radio-metal through the keto-enol moiety, resulting in positively charged 2:1 ligand to metal ratio complexes. The present study is focused on evaluating whether a suitable curcumin derivative that is more stably labeled with gallium-68 provides a radiotracer that preferentially accumulates in CRC lesions, and could be a useful early detection agent which enables nuclear medicine imaging techniques. CRC is globally the third most common type of cancer and is usually diagnosed by invasive techniques like sigmoidoscopy or colonoscopy, during which a colon specimen is collected. These procedures are then followed by medical imaging (usually CT scan) to determine if the disease has spread. While small polyps may be removed during colonoscopy, the cancerous state of large polyps may be assessed by a biopsy [25]. The introduction of a PET radiotracer that specifically accumulates in CRC might be useful for verifying the nature of these lesions without resort to invasive methods. Moreover, treatments used for colorectal cancer may include some combination of surgery, radiation therapy, chemotherapy and targeted therapy and nuclear medicine imaging may be a suitable technique to stage and assess follow-up of the patients.

In the present study, the structure of the first generation of gallium-68 curcumin complexes has been improved by adding DOTA, a strong gallium-68 chelator. DOTA was linked through an (amino)ethyl spacer to one of the phenol groups of the curcumin backbone. With the addition of DOTA, the new bio-conjugate (namely DOTA-C21) has an enhanced water solubility and a higher stability of the complex with gallium-68 is achieved in physiological conditions. Despite its solubility in water, ^1^H-NMR spectrum of DOTA-C21 in D_2_O (Appendix A) exhibited extremely broad signals in the aromatic region, suggesting that the hydrophobic backbone of curcumin interacted strongly by intramolecular π−π stacking. The signals could be sharpened by reducing the solvent polarity, for example, switching to methanol-d_4_, which reduced aggregation. The ^1^H-NMR spectrum in methanol-*d*_4_ (Appendix A) provided signals in the aromatic region attributable to the asymmetric curcumin moiety of DOTA-C21, while methylene signals of the ethyl spacer and DOTA were found in the aliphatic region (3–4.5 ppm). The structure of curcumin in DOTA-C21 is asymmetric due to the alkylation of one phenol group with the (amino)ethyl spacer, as clearly shown by two slightly shifted spin systems with equivalent spectral pattern (Appendix A). Complete assignment and chemical structure with atom numbering is reported in Section 4.1.1.

As shown in the ESI-LC-MS analysis (Appendix A), DOTA-C21 was present in two isomeric forms that could be assigned to the keto-enol and di-keto tautomer, respectively [17,18]. As the equilibrium was influenced by solvent polarity and pH, the ratio between the two forms could vary over the different analysis conditions. Due to the rapid interconversion between the two isomeric forms also in physiological conditions, it was not expected that the two isomers exhibited different in vivo properties but this assumption should be verified with further studies.

When Ga-DOTA-C21 complexes were synthesized mimicking the radiolabelling experimental conditions (95 °C, 30 min, L:M 1.5:1, 0.4 M NH_4_Ac), complex formation was confirmed by mass spectrometry and two compounds with the same *m*/*z*, corresponding to gallium-DOTA-C21 complexes at 1:1 metal to ligand molar ratio, were identified by ESI-LC/MS (Appendix A). The isotopic pattern (dashed square, Appendix A) highlighted the presence of gallium with its characteristic isotopic distribution in both compounds. Further LC/MS fragmentation experiments were carried out in order to exclude the involvement of the keto-enol moiety in the coordination sphere. As reported in Figure 3, the fragmentation pattern of the two isobaric peaks (panels B and C) were completely equivalent and could be attributed to cleavage of the curcumin structure, while gallium was always strongly bound by DOTA (Figure 10). This finding supported the hypothesis that the two isobaric peaks were due to the keto-enol and di-keto tautomers and not to different coordination modes.

To elucidate chelation mode and binding sites in depth, ^1^H-NMR titration of DOTA-C21 with Ga^3+^ was performed in methanol-d_4_ at 25 °C (Appendix A). The signals at 3.2 ppm (methylene groups of tetraazacyclododecane) and at 3.87 ppm (methylene groups in α to carboxylic groups) in the free ligand broadened and shifted at the first Ga^3 +^ addition, suggesting the involvement of the chelator in gallium coordination. A complete overview of 1:1 metal to ligand molar ratio complex formation for both proton and carbon resonances was shown by the overlap of ^1^H-,^13^C-HSQC-NMR spectra of the free ligand (DOTA-C21) and its gallium complex (Ga-DOTA-C21) (Figure 2). The metal coordination induces a de-shielding effect on all methylene groups of the chelator ring resulting in an increase of both proton and carbon chemical shifts. Particularly, metal complexation removes chemical shift equivalence of ring methylene protons, as clearly showed by ^1^H-,^1^H-COSY and ^1^H-,^13^C-HSQC-NMR spectra (Appendix A and Figure 4). Methylene group r_1_ shifts from ^1^H/^13^C δ (ppm) 3.35/49.77 to 4.00/54.65 and 3.38/54.65, respectively. A lower shift is observed for methylene group *r_2_* from ^1^H/^13^C δ (ppm) 3.26/49.64 to 3.47/57.11 and 3.41/57.11, respectively. These outcomes suggest the assignment of type r_1_ to those methylene groups in α position to carboxylic moiety directly involved in metal coordination (atom numbering 17, 18, 21 and 22). All atom numbering herein reported refers to Figure 11 and Figure 12), while *r_2_* type can be attributed to those methylene groups in α position to non-coordinating carboxylic (amidic) moiety (atom numbering 16, 19, 20 and 23). Methylene groups of the carboxylic arms in α to both nitrogen and carboxylic (amidic) moiety undergo the effect of metal coordination in a different way if the carboxylic group is involved in coordination or not. As shown in Figure 4, methylenes of the carboxylic arms shift from ^1^H/^13^C δ (ppm) 3.75–3.87/54.20 to 3.83/59.41, 3.79/60.64 and 3.74/61.68, respectively. Conversely, the chemical shifts of the ethyl spacer (H12 and H13) are unaffected by metal coordination.

Curcumin and curcuminoids act as fluorophores in the visible spectrum with strong solvent dependence, and consequently may be employed as fluorescent probes for distribution studies in cells and tissues. The typical main absorbance, due to the keto-enol group, was around 420 nm and the emission was around 500 nm, when excited at 420 nm wavelength [14]. In particular, the absorption of DOTA-C21 showed a maximum at 410 nm in water solution. When a titration with Ga^3 +^ was performed, no variations were observed at λ_max_ suggesting that the keto-enol moiety was not involved in gallium complexation. On the other hand, an increase in absorbance around 300 nm upon metal addition was detected, supporting that DOTA was the main moiety involved in metal chelation (Appendix A). The free ligand showed weak fluorescence emission upon excitation at λ_max_ (410 nm) with a maximum at 525 nm (Appendix A). These results differ from previously reported curcumin studies. In fact, while fluorescence quenching occurred due to gallium chelation by the keto-enol moiety [14], herein, emission spectra of DOTA-C21 were only slightly affected by gallium addition. This opposite behaviour furthermore confirmed the involvement of DOTA in gallium chelation rather than the keto-enol group, confirming what was seen by MS and NMR analyses.

Initially, DOTA-C21 was labelled with gallium-68 by preprocessing the generator elution by cation exchange purification [26]. This resulted in efficient labelling with high RCY and RCP after 5 min of incubation of just 10 nmol of precursor. In order to simplify the procedure, labelling was also attempted without preprocessing of the generator elutions. In these conditions, labelling was less effective, further purification of the final product was necessary due to the presence of high amounts of hydrolysed products, and 80 nmol of precursor was needed to drive the reaction to completion. This behavior could be ascribed to the presence of high levels of metal impurities that come from the generator and potentially compete with gallium-68 in the coordination with DOTA [19]. It is worth noting, that DOTA is known to have slow kinetics of complexation and requires harsh conditions for labelling. For these reasons, use of DOTA as a chelator for gallium-68 may enhance hydrolysis of the product and decomposition of the precursor. Use of chelators with smaller cavities such as NOTA and derivatives, or some acyclic chelators recently reported in literature such as HBED-CC or THP, would likely resolve these problems [27].

Similar to other gallium-68 labelled curcuminoid derivatives reported in literature [15] but differently from ^68^GaCl_3_, ^68^Ga-DOTA-C21 exhibited a time-dependent uptake in HT29 cells suggesting a relationship between the curcumin-like structure and the ability of the cells to bind and internalize the radiotracers. In particular, the addition of a specific chelator connected to a phenolic moiety appears to be advantageous for the internalization of the compound when compared to the our previously reported tracers where the keto-enol group was exploited to form gallium-68 complexes [15]. Once internalized or bound to the cell membrane, ^68^Ga-DOTA-C21 showed a slow efflux rate and 75% of the compound was still retained by the cells after 60 min of incubation. The mechanism by which curcumin or its derivatives are taken up in colorectal tumours has not been elucidated so far. It has been reported that curcumin has a structure suitable to bind nuclear vitamin D receptor and plays a role in colon cancer chemoprevention thanks to this property [28,29] but no specific experiments have been performed by the authors to confirm this finding. To test the hypothesis of VDR-mediated curcumin uptake, we analysed the VDR expression in HT29 cells and we found that this expression was higher than in other tumour cell lines and human lymphocytes. Then, to assess VDR-specificity, calcitriol, the hormonally active metabolite of vitamin D and VDR natural ligand, was used as a blocking agent. This would clarify the potential involvement of these receptors in ^68^Ga-DOTA-C21 uptake. Unfortunately, uptake of ^68^Ga-DOTA-C21 was not influenced by the presence of the excess of calcitriol (Figure 6C) therefore providing evidence that curcumin and calcitrol are not competitive substrates for the same binding site on VDR. In conclusion, seeing as this is a topic of great interest, further studies are necessary to evaluate the cellular uptake mechanism of curcumin and its radiolabelled derivatives. When the mechanism of accumulation will be elucidated, considerations about the structure-affinity of ^68^Ga-DOTA-C21 in comparison to other compounds will be possible.

The stability tests in whole blood (Figure 7) attested to a certain lability of the curcumin backbone that was probably partially metabolized into more polar products, which then underwent renal excretion in physiological conditions, as proved during the biodistribution study (kidney uptake 11.86 ± 1.1% ID/g at 1.5 h post injection). However, it was shown that 57 ± 3% of ^68^Ga-DOTA-C21 complexes remained intact after 120 min of incubation in human blood, comparing favourably with our other previously reported gallium-68 complexes [14], where the metal, coordinated by the keto-enol moiety, was rapidly released. Stability of ^68^Ga-DOTA-C21 in mouse blood appears comparable, if not superior, to previously reported F-18 labelled curcumin derivatives where the amount of unchanged radiotracer in rat plasma after 10 min post injection comprised only 45% of the total injected radioactivity [30]. ^68^Ga-DOTA-C21 exhibited a slow absorption rate into the main organs and a high amount of activity (12.73 ± 1.9% ID/g) was recorded in blood after 1.5 h post injection suggesting a strong binding interaction with serum albumin as reported for curcumin and their derivatives in general [30,31]. Significant uptake of ^68^Ga-DOTA-C21 occurred in the intestine (13.56 ± 3.3% ID/g), lungs (8.42 ± 0.8% ID/g), liver (5.81 ± 0.5% ID/g) and heart (4.70 ± 0.4% ID/g). ^68^Ga-DOTA-C21 rapidly accumulated in colorectal xenograft tumours with an uptake of 3.57 ± 0.3% ID/g after 1.5 h post injection. Compared to previously reported ^18^F-labelled curcumin derivatives, which exhibit fast clearance from the blood to the liver and fast hepatobiliary excretion to the intestine, ^68^Ga-DOTA-C21 blood clearance was slow and it showed both renal and hepatobiliary excretion. Generally, higher kidney and lower intestine uptake with respect to the fluorinated compounds were found. However, the amount of radiotracer accumulated in the intestine might still be a concern since it could mask the colorectal cancer uptake in a potential human examination. Liver uptake was comparable, indicating that ^68^Ga-DOTA-C21 remained partially metabolised, likely via hexahydrocurcumin-glucuroniside conjugation route. Similar to ^18^F-radiolabelled compounds, brain uptake was almost negligible [30,32].

## 4. Materials and Methods

### 4.1. General Procedures and Chemicals

All chemicals were reagent grade and used without further purification unless otherwise specified.

Complete information related to general laboratory procedures and instrumental analyses are reported in the Appendix A.

#### 4.1.1. Synthesis of 1,4,7,10-tetraazacyclododecane -1,4,7,10-tetraacetic (1,7-bis(4-hydroxy-3-methoxyphenyl)- 1,6-heptadiene-3,5-dione (DOTA-C21)

An anhydrous DMF (4 mL) solution of curcumin (100 mg, 0.27 mmol) and potassium carbonate (45 mg, 0.33 mmol) was added to 2-(Boc-amino)ethyl bromide (61 mg, 0.27 mmol). The mixture was stirred for 48 h at room temperature and then concentrated under reduced pressure. The residue was purified by flash silica-gel column chromatography (2% methanol/dichloromethane) to obtain compound **1** (16 mg, 12%) as a red-solid. ^1^H-NMR (600 MHz, CDCl_3_) δ 1.45 (s, 9H), 3.54–3.62 (m, 2H), 3.90 (s, 3H), 3.95 (s, 3H), 4.08–4.12 (m, 2H), 5.16–5.18 (m, 1H), 5.79 (s, 1H), 5.85 (s, 1H), 6.49 (dd, 2H, *J* = 6.0, 16.00 Hz), 6.92 (dd, 2H, *J* = 8.0, 15.6 Hz), 7.10 (ddd, 4H, *J* = 2.0, 5.6, 22.4 Hz), 7. 60 (d, 2H, *J* = 16.0 Hz).

An anhydrous dichloromethane (2.4 mL) solution of **1** (10.6 mg, 0.02 mmol) was added to TFA (0.6 mL). The mixture was sonicated for 20 min and then concentrated under reduced pressure. The residue was dissolved in DMF (3 mL) and diisopropylethylamine (21 μL, 0.12 mmol) and DOTA-NHS ester (19 mg, 0.025 mmol) were added. The mixture was stirred overnight at room temperature, and then concentrated under reduced pressure. The residue was purified by flash silica-gel column chromatography (20–40% acetonitrile/water containing 0.1% TFA) and freeze-dried to obtain **2** (10 mg, 38.5%) as a yellow-red colour solid. MS (ESI) *m*/*z* 798.4 [M + H]^+^.

^1^H-NMR (600 MHz, MeOD-d_4_) δ: 3.2–3.3 (H16→H23, m broad, 16H), 3.87 (H15/H24/H25/H26, m broad, 8H), 3.65 (H13, t, 2H), 3.93 (H11′, s, 3H), 3.94 (H11, s, 3H), 4.17 (H12, t, 2H), 6.00 (H1, s, 1H), 6.66 (H4′, d, 1H), 6.70 (H4, d, 1H), 6.85 (H9′, d, 1H), 7.05 (H9, d, 1H), 7.14 (H10′, dd, 1H), 7.22 (H10, dd, 1H), 7.24 (H6′, d, 1H), 7.29 (H6, d, 1H), 7.61 (H3′, d, 1H), 7.62 (H3, d, 1H). ^13^C-NMR (150.9 MHz, MeOD-d_4_) δ: 184.0 (C2′), 182.7 (C2), 150.1 (C8), 149.6 (C7), 149.2 (C8′), 148.0 (C7′), 141.1 (C3), 140.0 (C3′), 128.8 (C5), 127.1 (C5′), 122.8 (C10′), 122.5 (C10), 122.0 (C4), 120.8 (C4′), 115.2 (C9′), 113.0 (C9), 110.4 (C6/C6′), 100.7 (C1), 67.0 (C12), 55.2 (C11′), 55.1 (C11), 54.4 (C15), 54.1 (C24/C25/C26), 50.0 (C16→C23), 38.6 (C13). Atom numbering refers to Figure 11.

#### 4.1.2. Preparation of ^nat^Ga-DOTA-C21 Complexes

Compound **2** was dissolved in 0.4 M ammonium acetate (450 μg of ligand in 600 μL of buffer, pH 4.5) and stirred at 95 °C for 5 min. Then, 85 µL of 10 mM Ga(NO_3_)_3_·9H_2_O solution were added to the mixture in order to have a 1.5:1 metal to ligand molar ratio. The advancement of the complexation reaction was monitored by ESI-LC-MS. Reaction was completed after 30 min of heating (*m*/*z* 864.2–866.2). Complete data on chemical characterization of DOTA-C21 and its gallium complexes are reported in the Appendix A.

### 4.2. Radiolabelling of DOTA-C21 with Gallium-68

Postprocessing based labelling protocol: ^68^Ge/^68^Ga generator (EZAG, Berlin, Germany) was eluted with 5 mL 0.1 M HCl and gallium-68 was passed and fixed on a AG50W-X4, 200-400mesh hydrogen form, cartridge (Bio-Rad, Milan, Italy). Gallium-68 was eluted with 0.4 mL of a 97.56% acetone, 0.05 M HCl solution and an aliquot (80 MBq) was added to a vial containing 10 nmol of DOTA-C21 in a 0.2 M ammonium acetate solution (pH 4). The mixtures were incubated at 95 °C and kinetic studies were performed at 1, 3, 5, 10 and 20 min by TLC analysis.

Direct labelling protocol: ^68^Ge/^68^Ga generators were manually eluted with 4 mL of HCl solution (0.1 N or 0.05 N depending whether EZAG or ITG generator was used, respectively). 700 μL of eluates, containing around 200 MBq of gallium-68, were collected in a disposable vial containing 80 nmol of precursors and 65 μL of a 1.5 M sodium acetate in order to maintain the pH of the reaction around 4. The mixture was heated at 95 °C up to 10 min and consequently passed through a light C18 cartridge (Waters, Milan, Italy) to eliminate unlabelled gallium-68 and polar by-products. The cartridge was washed with 3 mL of saline and 1 mL of 10% ethanol solution and then eluted with 1 mL of 95% ethanol solution followed by 2 mL water. Aliquots of the crude mixture and of the final product after purification were collected to assess the radiochemical purity of the product (RCP) by UHPLC analysis. Every preparation was performed at least in triplicate and all the incorporation yields were computed by considering the radiochemical purity (RCP) obtained from the UHPLC or TLC analyses.

### 4.3. In Vitro Stability Studies

For assessing the stability, aliquots of ^68^Ga-DOTA-C21 solution (1 mL, approx. 5 nmol, 37 MBq) were incubated with 1 mL of i) PBS (0.2M pH = 7.2), ii) human serum (HS), iii) human whole blood (HB) at 37 °C for different time points (10, 40, 70, 120 min). Samples incubated with HB were centrifuged at 3000 rpm for 10 min to precipitate the blood cells and a solution (200 μL) of ACN/H_2_O/TFA 50/45/5 *v*/*v*/*v* was added to 400 μL of the supernatant. Samples incubated with HS were treated only with ACN/H_2_O/TFA 50/45/5 *v*/*v*/*v* solution. After another centrifugation under the same conditions, the supernatant was injected into an UPLC for assessing the stability of the preparation.

### 4.4. Cell Culture and Animal Models

HT29 (colorectal adenocarcinoma) cells were kindly provided by Dr Alessandro Zerbini from the Unit of Infectious Diseases and Hepatology, Azienda Ospedaliero-Universitaria di Parma (Parma, Italy). The cells were grown in DMEM + 10% FBS supplemented with penicillin and streptomycin at 37 °C in a 5% CO_2_ incubator. To determine protein yield, cells were lysed with a radio immune-precipitation assay (RIPA) buffer (Santa Cruz Biotechnology) and protein concentration was determined with the detergent compatible (DC) protein assay (Bio-Rad) following the manufacturer’s instructions using bovine serum albumin as protein standard.

All animal experiments were performed in compliance with the guidelines for the care and use of research animals established by The University of Texas MD Anderson Cancer Center (ethic approval number 1179). Mice were maintained in sterile conditions and could eat and drink ad libitum. Mice were housed in a 12 h light-dark cycle. Adult female mice (athymic nude, Taconic Biosciences, 7–8 weeks old) were injected subcutaneously with 9 × 10^6^ HT29 cells in 50% matrigel on the right flank.

### 4.5. In Vitro Uptake, Internalization and Efflux in Colorectal Cancer Cell Line

Cellular uptake, internalization, blocking and efflux of ^68^Ga-DOTA-C21 were studied in HT29 colon carcinoma cells. All experiments were performed in triplicate (unless otherwise stated). For uptake studies, HT29 cells were incubated at 37 °C with 20 µL (ca. 2 MBq, 0.2 nmol, 0.1 µM final concentration) of ^68^Ga-DOTA-C21 and ^68^GaCl_3_ as a negative control. For blocking experiments HT29 cells were preincubated for 1 h at 37 °C with 100 μL of 1 mg/mL calcitriol solution and then with 20 µL (ca. 2 MBq, 0.2 nmol, 0.1 µM final concentration) of ^68^Ga-DOTA-C21. Uptake was stopped after 5, 10, 30 and 60 min by removing the medium and the cells were washed twice with ice-cold PBS or 0.1 M (pH 2.9) ice-cold glycine solution for discriminating between the total bound activity and the internalized activity, respectively. Finally, the cells were detached with 2 mL of trypsin/EDTA 0.25% solution at 37 °C and centrifuged in order to separate the supernatant from the cells pellet. The radioactivity associated to the pellets was measured in the dose calibrator and normalized for the protein contents. For efflux studies, cells were incubated with 20 µL (ca. 2 MBq, 0.2 nmol, 0.1 µM final concentration) of ^68^Ga-DOTA-C21 for 1 h at 37 °C. The uptake was stopped by removing the supernatant and cells were washed twice with ice-cold PBS. Then 1 mL of fresh prewarmed (37 °C) medium was added and cells were newly incubated at 37 °C for 5, 10, 30, 60 min. After these time points, the media were collected and the radioactivity released from the cells was measured in the dose calibrator.

### 4.6. Biodistribution of ^68^Ga-DOTA-C21

^68^Ga-DOTA-C21 biodistribution was determined in mice bearing HT29 subcutaneous tumour model (*n* = 5) at 1.5 h post tail vein injection of 3.7 MBq (1 mL, 0.2 µM solution) of radiotracer. Blood samples were drawn from the right leg using a femoral vein puncture as well as an intra-cardiac puncture to draw blood from the heart to assess the in vivo stability of the radiotracer. The animals were euthanized using 2% isoflurane and exsanguinated, and the thoracic cavity opened. Organs were excised, washed with saline, dried with absorbent tissue paper, counted on a gamma-counter (Packard BioScience Cobra II Auto-Gamma, Meriden, USA) and weighed. Organs of interest included: tumour, heart, spleen, lungs, liver, kidneys, stomach, small intestine, large intestine, muscle, bone and brain. The uptake in each organ was calculated as a percentage of the injected dose per gram of tissue (% ID/g). Blood samples were immediately centrifuged to precipitate the blood cells and a solution (200 μL) of ACN/H_2_O/TFA 50/45/5 *v*/*v*/*v* was added to 400 μL of the supernatant. After a further centrifugation, the extracted solution was injected into an HPLC.

### 4.7. PET Imaging of Tumour Bearing Mice and Analysis

Mice were briefly anaesthetized (<5 min) using 1% to 2% isoflurane with O_2_ as a carrier. Mice were injected i.v. or i.p. with ^68^Ga-DOTA-C21 in sterile phosphate-buffered saline (PBS) (10% EtOH or 50% EtOH respectively) with a target of 3.7 MBq (1 mL, 0.2 µM solution) per mouse. Actual injected dose was calculated based on measuring the pre and postinjection activity in the syringe with a dose calibrator (Capintec, Florham Park, NJ, USA). Mice were then returned to their cages, quickly became ambulatory and could move freely, eat and drink ad libitum for ~45 min. Mice were then anaesthetized using 1% and 3% isoflurane, transferred to a preclinical PET/SPECT/CT system (Albira PET/SPECT/CT, Bruker, Ettlingen, Germany) and maintained at 0.5% to 2% isoflurane with continuous monitoring of respiration during the acquisition. PET images were acquired for 10 min using a 15 cm FOV centred on the tumour; CT images were acquired for fusion using a 7 cm FOV also centred on the tumour. The same procedure was repeated for the 1h and 2 h PET/CT scan. The 10 min PET/CT dynamic scan was recorded immediately after injection of the tracer, and then mice were allowed to awake and freely move around their cages until the 1 h time point. Images were reconstructed using MLEM and FBP for PET and CT, respectively, and automatically fused by the software. Image data were decay corrected to injection time (Albira, Bruker, Ettlingen, Germany) and expressed as % ID/cc (PMOD, PMOD Technologies). Tumour-to-muscle ratios (T/M) were calculated by dividing the activity present in the tumour by the activity present in the muscle.

### 4.8. Statistical Analysis

Student’s *t*-test was used to determine whether there were any statistically significant differences between the means of two independent (unrelated) groups. The threshold for statistical significance was set at *p* < 0.05.

## 5. Conclusions

The new DOTA derivative of curcumin (DOTA-C21) displayed coordinating property typical of DOTA structure, with the involvement of nitrogen atoms of the rings and two carboxylic arms, as demonstrated by NMR and MS analysis. Gallium-68 labelling conditions are harsh (95 °C, 5′), however they provide good RCY and RCP in spite of the methods used for the reaction (prepurification of the eluate or direct labelling and postpurification). Despite high instability of curcumin in physiological conditions, stability of ^68^Ga-DOTA-C21 in mouse blood appears comparable, if not superior, to previously reported ^18^F-labelled curcumin derivatives, and these data are compatible with diagnostic applications. ^68^Ga-DOTA-C21 rapidly accumulated in colorectal xenograft tumours but after 1.5 h post injection, exhibits a slow absorption rate into the main organs and a high amount of activity in blood. Unfortunately, the high uptake in blood, liver and intestines prevents the direct use of this derivatives as an imaging agent for colorectal cancer. However, further studies are needed to understand the mechanism of accumulation of ^68^Ga-DOTA-C21 and to design strategies to increase its selectivity for colorectal tumour since the molecular structure of this radiotracer is an interesting foundation to develop further compounds with improved stability and pharmacokinetics.

## Figures and Tables

**Figure 1 molecules-24-00644-f001:**
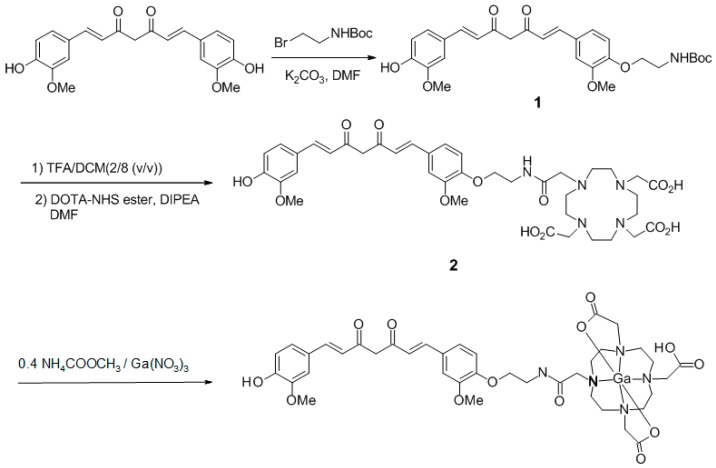
Synthetic pathway for the preparation of DOTA-C21 and its complexation with Ga(III) by reacting DOTA-C21 with Ga(NO_3_)_3_ at 1.5:1 metal-to-ligand molar ratio in 0.4 M ammonium acetate solution for 30 min at 95 °C.

**Figure 2 molecules-24-00644-f002:**
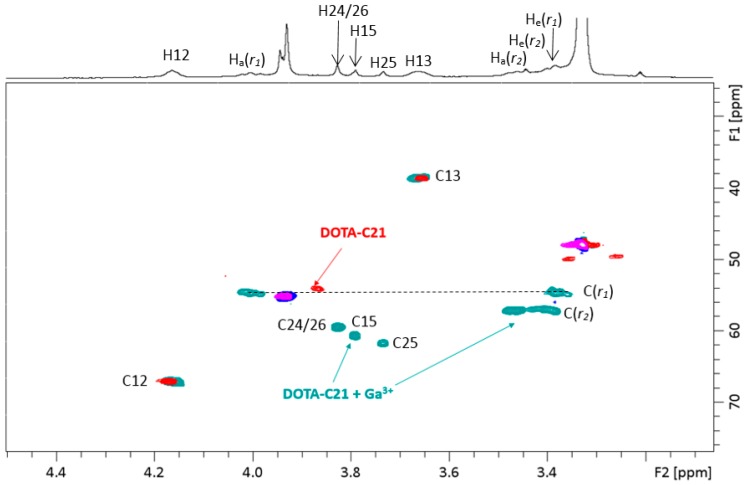
Aliphatic region overlap of ^1^H-, ^13^C-HSQC-NMR spectra of DOTA-C21 (red (CH_2_) and purple (CH/CH_3_)) and Ga-DOTA-C21 (green (CH_2_) and blue (CH/CH_3_)). All the spectra were acquired in MeOD-*d*_4_ at 25 °C ([DOTA-C21] = 0.63 mM). Atom numbering refers to Figures 11 and 12.

**Figure 3 molecules-24-00644-f003:**
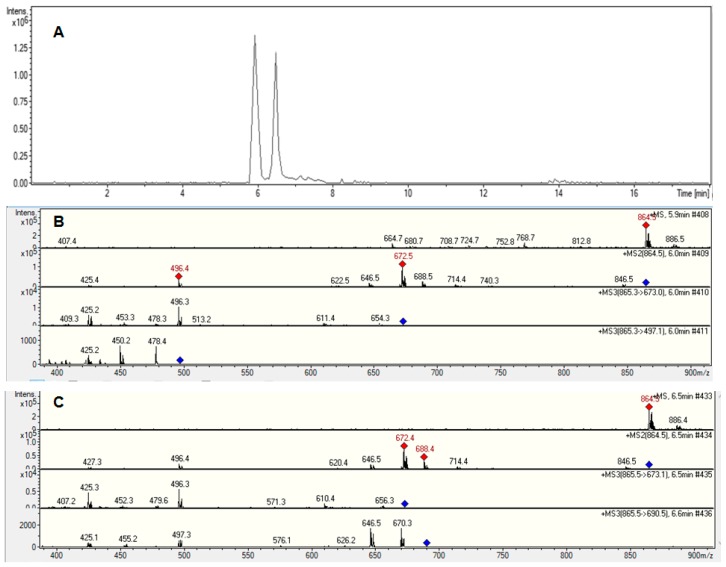
LC-MS/MS fragmentation experiments on Ga-DOTA-C21 complexes with *m/z* 864.5 [M + H]^+^. HPLC-MS chromatogram (**A**). Fragmentation pathway of the *m/z* 864.5 [M + H]^+^ ion corresponding to the peak with retention time 5.9 min (**B**). Fragmentation pathway of the *m/z* 864.5 [M + H]^+^ ion corresponding to the peak with retention time 6.5 min (**C**).

**Figure 4 molecules-24-00644-f004:**
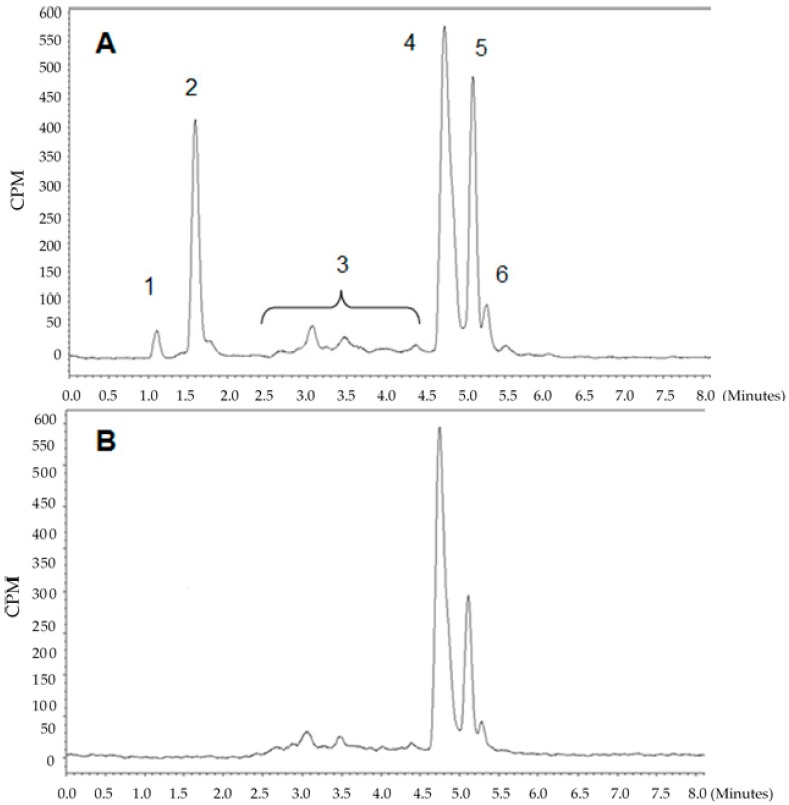
Representative UHPLC chromatograms (radiochemical detector) of a ^68^Ga-DOTA-C21 preparation before (**A**) and after (**B**) SPE purification. (1) ^68^Ga-free, (2) ^68^Ga-hydrolyzed products, (3) ^68^Ga-labelled byproducts, (4) and (5) ^68^Ga-DOTA-C21 isomers, (6) ^68^Ga-labelled degradation product.

**Figure 5 molecules-24-00644-f005:**
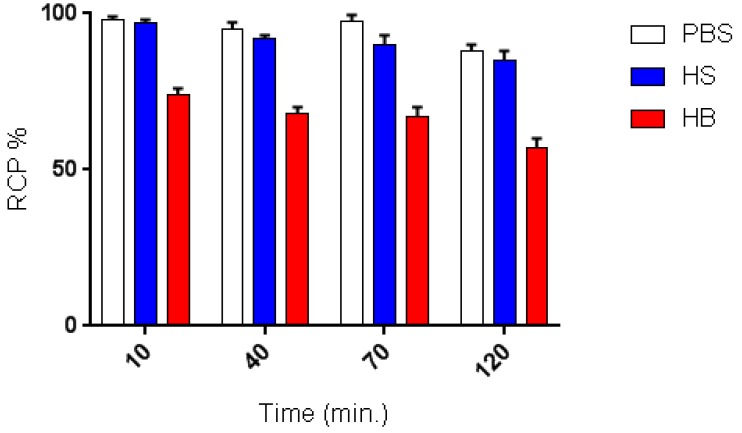
Stability of ^68^Ga-DOTA-C21 complex incubated over time (10, 40, 70, 120 min.) in different media (*n* = 3, mean ± SD).

**Figure 6 molecules-24-00644-f006:**
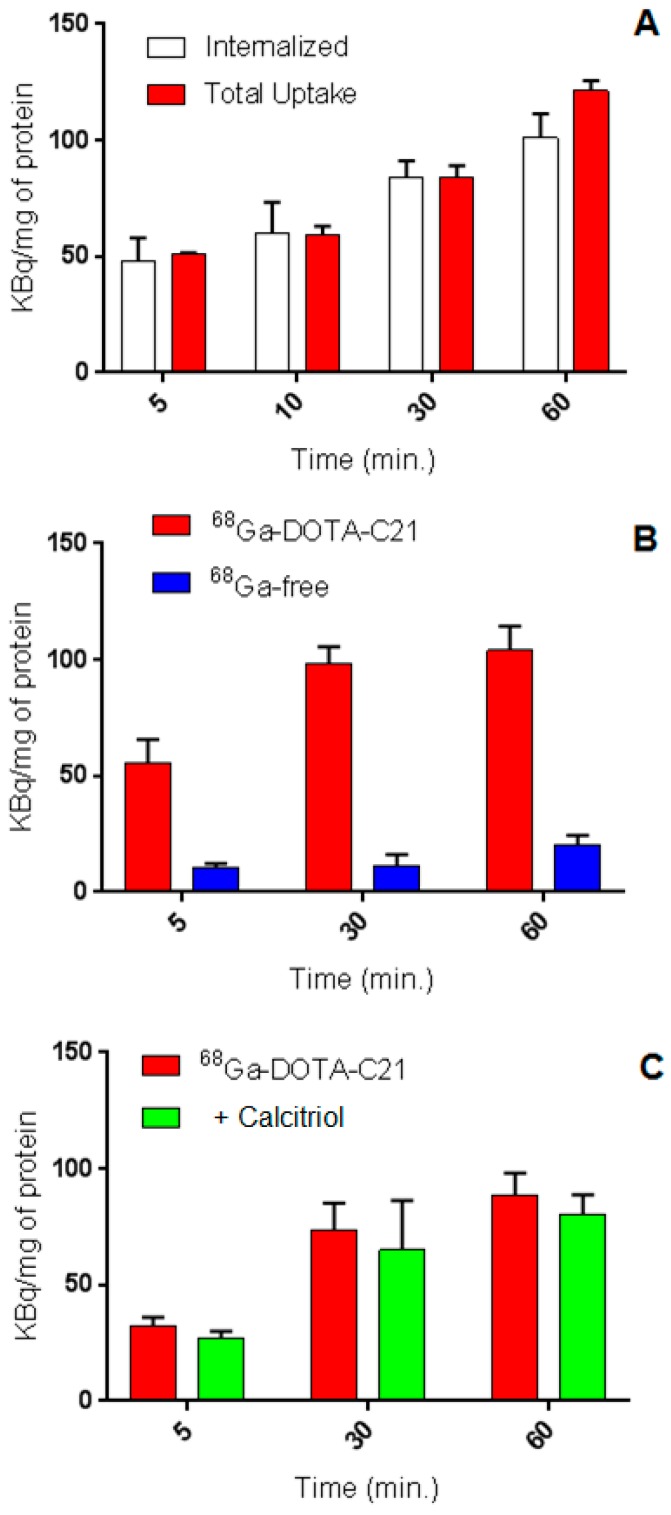
Uptake and internalization at different time points of ^68^Ga-DOTA-C21 in HT-29 colorectal cancer cell line (*n* = 3, mean ± SD) (**A**). Comparison between the uptake of ^68^Ga-DOTA-C21 and ^68^GaCl_3_ in HT-29 cells indicating curcumin structure dependent accumulation (*n* = 3, mean ± SD) (**B**). Comparison between the uptake of ^68^Ga-DOTA-C21 and ^68^Ga-DOTA-C21 + calcitriol in HT-29 cells (*n* = 3, mean ± SD) (**C**).

**Figure 7 molecules-24-00644-f007:**
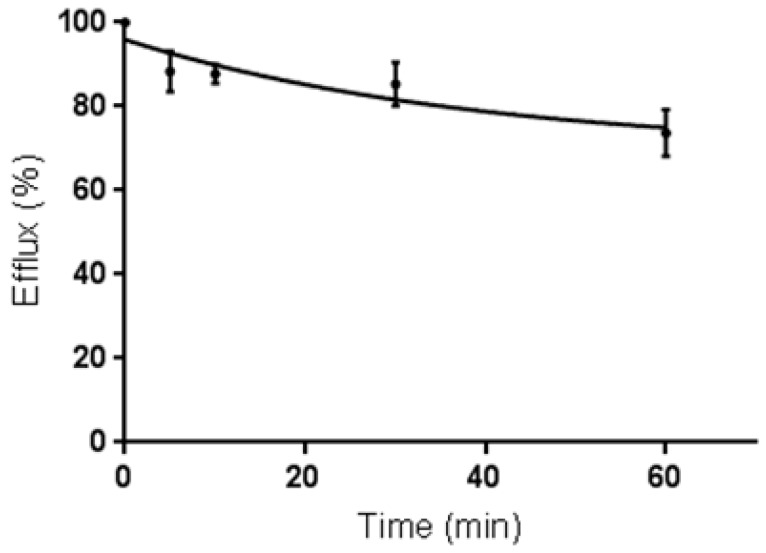
Externalization of ^68^Ga-DOTA-C21 in HT-29 colorectal cancer cell line at different time points (*n* = 3, mean ± SD).

**Figure 8 molecules-24-00644-f008:**
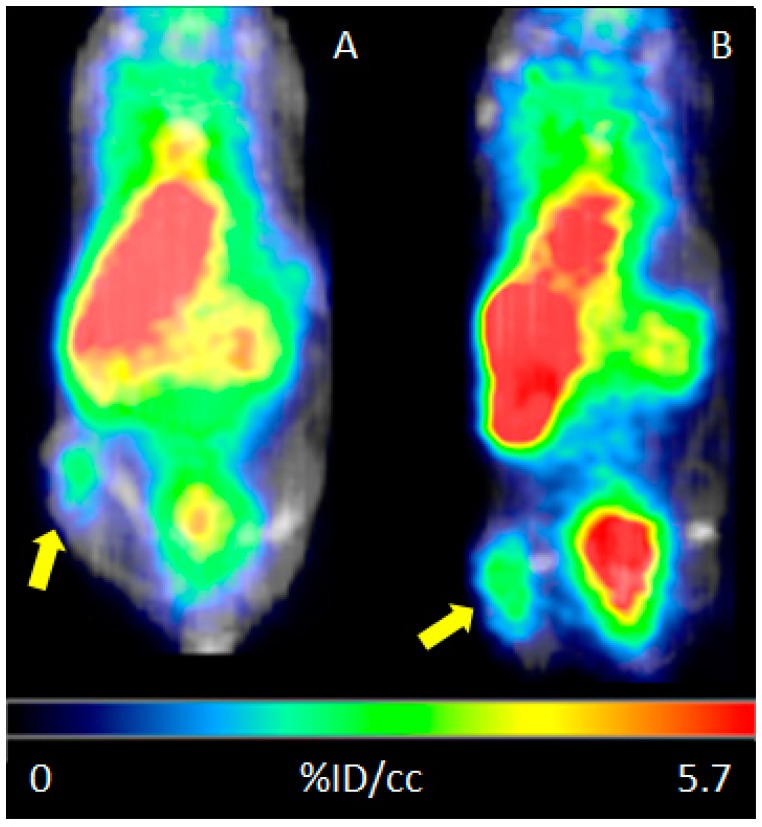
Representative microcoronal-PET/CT scans of HT29 colorectal tumour bearing mouse at 60 min (**A**) and 120 min (**B**) post i.p. injection of ca. 37 MBq of ^68^Ga-DOTA-C21.

**Figure 9 molecules-24-00644-f009:**
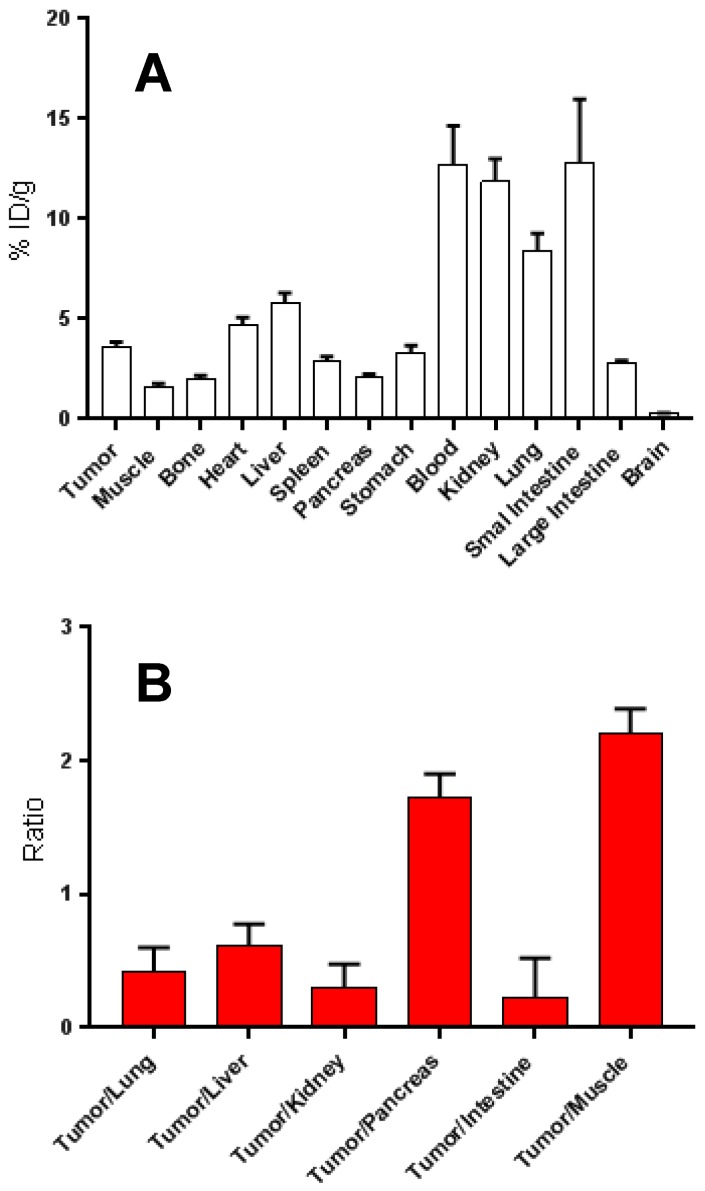
Biodistribution in multiple organs of ^68^Ga-DOTA-C21 in HT-29 colorectal cancer bearing nude mice (*n* = 5, mean ± SD) at 90 min post i.v. injection. (**A**). Ratios of the tumour uptake to major organs (*n* = 5, mean ± SD) for ^68^Ga-DOTA-C21 at 90 min post i.v. injection (**B**).

**Figure 10 molecules-24-00644-f010:**
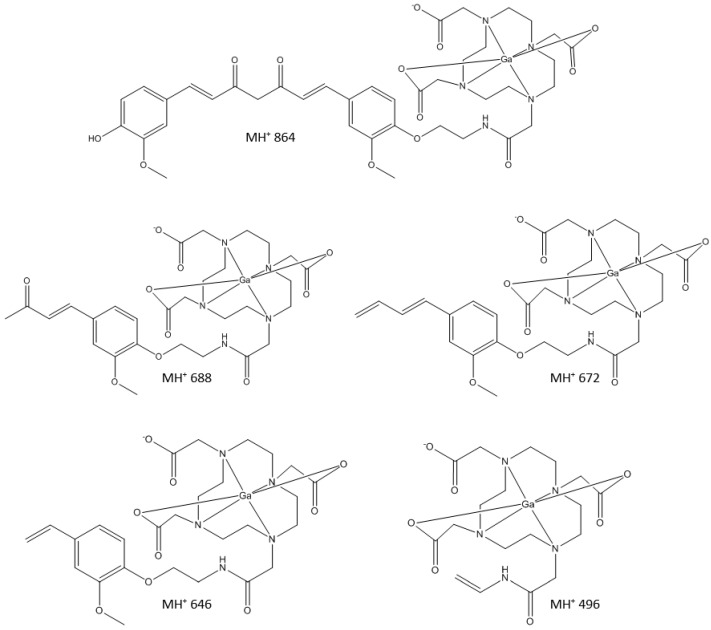
Molecular fragments of Ga-DOTA-C21 attributed to the correspondent *m*/*z* signals detected in panel B and C of Figure 3.

**Figure 11 molecules-24-00644-f011:**
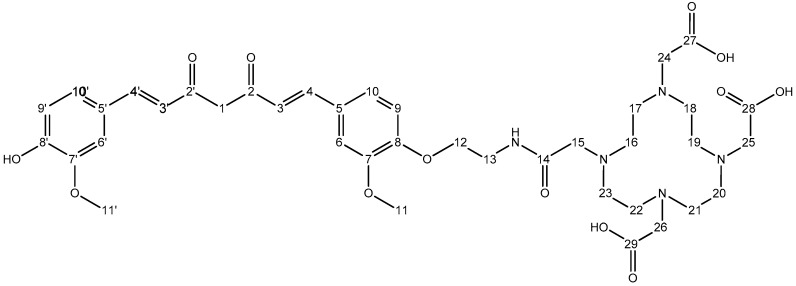
Chemical structure of DOTA-C21 with atom numbering.

**Figure 12 molecules-24-00644-f012:**
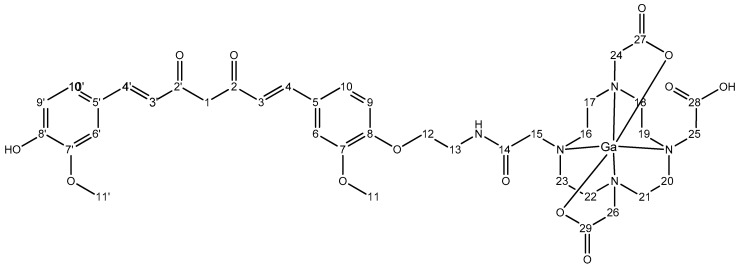
Chemical structure of Ga-DOTA-C21 with atom numbering.

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
