# Peer review of "Development of a Potential Gallium-68-Labelled Radiotracer Based on DOTA-Curcumin for Colon-Rectal Carcinoma: From Synthesis to In Vivo Studies"

_molecules, 2019, doi:10.3390/molecules24030644_

Round 1
Reviewer 1 Report
The manuscript by Giulia Orteca et al. submitted to Molecules reports a novel gallium-68 labelled curcumin derivative (68Ga-DOTA-C21) as a potential radiotracer for early detection of colorectal cancer. The research field is about the development of molecular imaging tracers. In general, the paper is well written. The experimental methods are described comprehensively and the interpretations and conclusions are justified by the results. This work is original and is suitable in content for Molecules. However, it is in need of some minor revisions as suggested below:
1. On Page 10, line 237, “60 minutes” should be “at 60 minutes”.
2. On Page 14, line 352, “1.5 hours post injection” should be “at 1.5 hours post injection”.
3. The authors should afford the comparison of the biodistribution of 68Ga-DOTA-C21 and other gallium-68 labelled curcuminoid complexes and 18F labelled curcuminoid complexes.
Author Response
Estimated Reviewers,
the Authors want to thank You for the careful reading, the reviewing efforts, and the helpful suggestions about the manuscript “Development of a potential gallium-68 labelled radiotracer based on DOTA-curcumin for colon-rectal carcinoma: from synthesis to in vivo studies”. The Authors endeavoured to address all Your comments in the most clear and exhaustive way. In the point-by-point reply, the Authors’ replies to Your comments are written in red and any addition/change to the original manuscript is highlighted in yellow in the revised version in order to be evident.
Thanking in advance for your reviewing efforts, the Authors hope that the revised submission could deem the publication in the Special Issue “New Trends in Production and Applications of Metal Radionuclides for Nuclear Medicine” in Molecules.
Best Regards
Erika Ferrari (On behalf of all Authors)
Point-by-point reply:
1. On Page 10, line 237, “60 minutes” should be “at 60 minutes”.
The text has been edited as the Reviewer suggested.
2. On Page 14, line 352, “1.5 hours post injection” should be “at 1.5 hours post injection”.
The text has been edited as the Reviewer suggested.
3. The authors should afford the comparison of the biodistribution of 68Ga-DOTA-C21 and other gallium-68 labelled curcuminoid complexes and 18F labelled curcuminoid complexes.
To the authors’ knowledge, biodistribution of other gallium-68 labelled curcuminoid complexes has not been reported in the literature so far. Comparison between 68Ga-DOTA-C21 and the only 18F-labelled curcuminoids whose distribution is reported in literature is afforded in the discussion section from line 387 to 395 of the revised version of the manuscript.
Reviewer 2 Report
The authors synthesized and characterized a precursor and a non-radioactive complex. The in vitro stability, cell uptake, internalization and efflux properties of the probe were studied in HT29 cells, and the in vivo targeting ability and biodistribution were investigated in mice. The manuscript can be accepted for publication in Molecules, after minor revision. The authors should revise the manuscript according to the following comments.
1. The resolution of the figures 10, 11 and 12 should be increased
2. The authors should give the cell line origin of HeLa, MCF-7, HT29, and HCT-116 in the introduction part.
3. The absorbance of UV spectrum (Figure S10) is too low, the absorbance should be at least 1.
4. The authors are not mentioning in the text the concentration of the compound for uptake, internalization and efflux tests. Moreover, in which concentration the compounds is toxic against HT29 cells
5. The authors should also mention in which concentration of the compound the mice were injected
6. In the figures S9 and S10 it is not clear what indicated by the different spectrum colour. Please explain
7. In the figure S12, the axes of the diagrams are illegible
Author Response
Estimated Reviewer,
the Authors want to thank You for the careful reading, the reviewing efforts, and the helpful suggestions about the manuscript “Development of a potential gallium-68 labelled radiotracer based on DOTA-curcumin for colon-rectal carcinoma: from synthesis to in vivo studies”. The Authors endeavoured to address all Your comments in the most clear and exhaustive way. In the point-by-point reply, the Authors’ replies to Your comments are written in red and any addition/change to the original manuscript is highlighted in yellow in the revised version in order to be evident.
Thanking in advance for your reviewing efforts, the Authors hope that the revised submission could deem the publication in the Special Issue “New Trends in Production and Applications of Metal Radionuclides for Nuclear Medicine” in Molecules.
Best Regards
Erika Ferrari (On behalf of all Authors)
Point-by-point reply:
1. The resolution of the figures 10, 11 and 12 should be increased
As suggested, the resolution of figures was improved.
2. The authors should give the cell line origin of HeLa, MCF-7, HT29, and HCT-116 in the introduction part.
The proper references has been added to the revised form of the manuscript in which it is possible to find the origin of the cell lines used in that studies.
3. The absorbance of UV spectrum (Figure S10) is too low, the absorbance should be at least 1.
The authors partly agree with this comment of the Reviewer since the values of absorbance reported in Fig. S10 are quite low but they still give the information needed. The reason of working with a so diluted solution is due to the few amount of the investigated compound. For this reason, the authors decided to use the same solution for both UV-vis and fluorescence spectroscopies analyses and had to work with diluted solutions otherwise fluorescence could have been prevented by signal saturation.
4. The authors are not mentioning in the text the concentration of the compound for uptake, internalization and efflux tests. Moreover, in which concentration the compounds is toxic against HT29 cells.
The concentrations have now been added to the text in the “materials and methods” paragraph (4.5 In vitro uptake, internalization and efflux in colorectal cancer cell line) as requested by the Reviewer. The authors have not tested at which concentration DOTA-C21 is toxic for HT29 cells, however the lead compound curcumin has an IC50 value close to 13 mM for HT29 cell line (BMC Cancer. 2009; 9: 99, doi: 10.1186/1471-2407-9-99). This value is more than 2 magnitude order greater than the concentration used for the in vitro/in vivo studies with radioactive compounds. Hence, the authors believe toxicity is not a concern for this study.
5. The authors should also mention in which concentration of the compound the mice were injected
The concentrations have now been added to the manuscript in the “materials and methods” paragraph 4.6 and 4.7.
6. In the figures S9 and S10 it is not clear what indicated by the different spectrum colour. Please explain
A legend with colour assignment was added to both figures, as suggested.
7. In the figure S12, the axes of the diagrams are illegible
The Author apologize for the poor resolution of Fig. S12 axes, a new picture is provided in the revised manuscript.
Reviewer 3 Report
The manuscript “Development of a potential gallium-68 labelled radiotracer based on DOTA-curcumin for colon-rectal carcinoma: from synthesis to in vivo studies” submitted by Orteca et al. describes the synthesis and characterization of the novel curcumin based tracer 68Ga-DOTA-C21. The compound has been well characterized using suitable NMR and MS techniques. Synthesis and characterization have been described very well in the manuscript. However, the preclinical evaluation of the compound seems insufficient. Further, most of the preclinical results are not discussed. Regarding the preclinical experiments, the referee criticizes the following points:
· The plasma protein binding of the compound was not investigated. This should have been conducted before the in vivo experiments. In line 360 the authors even state that curcumins are known to bind towards serum albumin.
· In the cell binding and uptake experiments, calcitriol is used as blocking substance. However, the blocking seems ineffective. Instead of blocking with calcitriol the authors could have attempted to block the tracer with free curcurmin like in the previous study (ref. 15). As consequence, neither the specificity of the uptake, nor the unspecific uptake is known. Without subtraction of the unspecific binding, the 83 % “internalization” (line 172) are meaningless. It might as well be unspecific binding/uptake – which would make sense regarding the exceptional high uptake.
With respect to the above mentioned flaws in the characterization of the compound conduction of the in vivo experiment is not sufficiently justified. It is impossible to know whether the uptake is specific or mediated via an unknown unspecific pathway. Therefore, the results of the in vivo experiment seem meaningless and the referee has to raise ethical concerns. In the current form the manuscript unfortunately does not justify publication in Molecules MDPI.
Anyway, the chemical part of the manuscript is written very well and deserves publication in a suitable journal. Repetition of the in vitro evaluation of the compound respecting the above mentioned points should be considered.
Author Response
Estimated Reviewer,
the Authors want to thank You for the careful reading, the reviewing efforts, and the helpful suggestions about the manuscript “Development of a potential gallium-68 labelled radiotracer based on DOTA-curcumin for colon-rectal carcinoma: from synthesis to in vivo studies”. The Authors endeavoured to address all Your comments in the most clear and exhaustive way. In the point-by-point reply, the Authors’ replies to Your comments are written in red and any addition/change to the original manuscript is highlighted in yellow in the revised version in order to be evident.
Thanking in advance for your reviewing efforts, the Authors hope that the revised submission could deem the publication in the Special Issue “New Trends in Production and Applications of Metal Radionuclides for Nuclear Medicine” in Molecules.
Best Regards
Erika Ferrari (On behalf of all Authors)
Point-by-point reply:
The manuscript “Development of a potential gallium-68 labelled radiotracer based on DOTA-curcumin for colon-rectal carcinoma: from synthesis to in vivo studies” submitted by Orteca et al. describes the synthesis and characterization of the novel curcumin based tracer 68Ga-DOTA-C21. The compound has been well characterized using suitable NMR and MS techniques. Synthesis and characterization have been described very well in the manuscript. However, the preclinical evaluation of the compound seems insufficient. Further, most of the preclinical results are not discussed. Regarding the preclinical experiments, the referee criticizes the following points:
The Authors thank the Reviewer for the positive comments about the synthetic and analytical part of the manuscript but only partially agree with the comments upon the preclinical studies and the Reviewer’s conclusions (please, see the point-by-point reply). However, the Authors agree with the Reviewer that in vitro results should be better discussed. Following the Reviewer suggestion, the discussion has been improved in the new version of the manuscript and lines 349 -356, lines 262-364 and lines 366-373 have been added.
· The plasma protein binding of the compound was not investigated. This should have been conducted before the in vivo experiments. In line 360 the authors even state that curcumins are known to bind towards serum albumin.
High binding of the curcuminoid-like structures to plasma proteins is well established in the literature, therefore the authors deem further experiments un-necessary to confirm these results. Notably, - as part of a pilot validation of a new tracer like the one presented in this manuscript - the authors considered the in vivo biodistribution of paramount importance since it affords real insight into the radiotracer behaviour in vivo. In addition, in reference 30, in vivo imaging of a labelled curcuminoid derivative is provided but neither stability or protein binding studies are reported.
In the cell binding and uptake experiments, calcitriol is used as blocking substance. However, the blocking seems ineffective. Instead of blocking with calcitriol the authors could have attempted to block the tracer with free curcumin like in the previous study (ref. 15). As consequence, neither the specificity of the uptake, nor the unspecific uptake is known. Without subtraction of the unspecific binding, the 83 % “internalization” (line 172) are meaningless. It might as well be unspecific binding/uptake – which would make sense regarding the exceptional high uptake.
As reported in lines 178 – 184 and 373 – 380 of the old version of the manuscript (lines 178-184 and 357-367 of the new version), calcitriol was not used as general blocking substance but specifically for testing if the 68Ga-DOTA-C21 uptake was mediated by the VDR receptors. Since the mechanism of curcumin and curcumin derivatives high uptake in colon rectal carcinoma is not yet elucidated, this experiment was used as an attempt to investigate whether VDRs were involved. On the other hand, experiments of blocking with a curcumin excess were not performed because authors believe that this kind of experiments are meaningful only once the mechanism of uptake is clarified. Clarifying the mechanism of curcumin accumulation in tumours is beyond the scope of this submitted manuscript. To show that the uptake in HT29 cells is curcuminoid dependent (in spite of the mechanism involved), the uptake of 68Ga-DOTA-C21 was compared to that of 68GaCl3 and the differences are evidently robust (Figure 6B). Honestly, authors do not understand what the Reviewer refers citing ref. 15 since in that publication the cell uptake was performed exactly in the same way described herein and no blocking study with curcumin was performed. The author’s point of view and the reason of using calcitriol, are now better described in the Discussions from line 357-370.
With respect to the above mentioned flaws in the characterization of the compound conduction of the in vivo experiment is not sufficiently justified. It is impossible to know whether the uptake is specific or mediated via an unknown unspecific pathway. Therefore, the results of the in vivo experiment seem meaningless and the referee has to raise ethical concerns. In the current form the manuscript unfortunately does not justify publication in Molecules MDPI.
The authors acknowledge that the in vitro results were not as robust as expected, however, this does not negate evaluating the radiotracer in vivo since the in vivo pharmacokinetics, tumour microenvironment, radiotracer stability, tumour growth and organization in a living system is completely different from a cell culture media or from a blood sample. It is so quite restrictive to state that the in vivo experiments are not justified on the basis of alleged inaccuracies and incompletion of the in vitro parts since only an in vivo evaluation could give significant insights on a radiotracer behaviour and on its transferability into human being in spite of in vitro findings. Finally, authors believe that, for the purposes of the manuscript and at the present level of study on the topic, our focus was not to understand the specific mechanism of cellular uptake of 68Ga-DOTA-C21 but rather demonstrate in a pilot in vivo experiment the potential of our radiotracer to image CRC. While the authors agree that the specific uptake mechanism of curcumin and its derivatives is vital to the field and translation of these compounds into the clinic, this is beyond the scope of the current manuscript and requires further future studies.
Anyway, the chemical part of the manuscript is written very well and deserves publication in a suitable journal. Repetition of the in vitro evaluation of the compound respecting the above mentioned points should be considered.
The authors thank the Reviewer for the considerations and believe that Molecules is a suitable platform for this manuscript since it perfectly fits the scope of the journal and in particular of the special issue “New Trends in Production and Applications of Metal Radionuclides for Nuclear Medicine”. While the authors greatly appreciate the Reviewer’s experimental suggestions, we recognize that the manuscript has some unfavourable in vitro results but politely ask the Reviewer to consider that what is observed in vitro does not always dictate how a compound will behave in vivo. Furthermore, the authors would like to emphasize that the in vivo study was a pilot experiment, as stated in the Conclusion of the manuscript, 68Ga-DOTA-C21 compound is only a step (the first) in the translation of this compound into the clinic.
Reviewer 4 Report
A manuscript by Orteca G et al describing development of a potential gallium-68 labelled radiotracer based on DOTA-curcumin for colorectal carcinoma is very interesting and well written. In my opinion it deserves publishing because of the novelty of design research and very high quality of the presentation of obtained results.
Author Response
Estimated Reviewer,
the Authors want to thank You for the careful reading, the reviewing efforts, and the helpful suggestions about the manuscript “Development of a potential gallium-68 labelled radiotracer based on DOTA-curcumin for colon-rectal carcinoma: from synthesis to in vivo studies”. The Authors endeavoured to address all Your comments in the most clear and exhaustive way. In the point-by-point reply, the Authors’ replies to Your comments are written in red and any addition/change to the original manuscript is highlighted in yellow in the revised version in order to be evident.
Thanking in advance for your reviewing efforts, the Authors hope that the revised submission could deem the publication in the Special Issue “New Trends in Production and Applications of Metal Radionuclides for Nuclear Medicine” in Molecules.
Best Regards
Erika Ferrari (On behalf of all Authors)
Point-by-point reply:
A manuscript by Orteca G et al describing development of a potential gallium-68 labelled radiotracer based on DOTA-curcumin for colorectal carcinoma is very interesting and well written. In my opinion it deserves publishing because of the novelty of design research and very high quality of the presentation of obtained results.
The Authors thank the Reviewer for the positive comments.
Round 2
Reviewer 3 Report
The referee acknowledges the reply by the authors regarding the in vitro and in vivo experiments (points 2-5 in the reply), however, with all respect, does not agree on the opinion of the authors. In detail:
Most importantly, the unspecific binding was not assed during in vitro experiments. This is even confirmed by the authors in the reply (paragraph 3 of the reply: “On the other hand, experiments of blocking with a curcumin excess were not performed because authors believe that this kind of experiments are meaningful only once the mechanism of uptake is clarified.”). However, without this vital information it remains totally unclear, if the observed uptake in mouse is specific or unspecific. Therefore, the information from this animal experiment cannot be interpreted correctly, a fact which the referee considers as unethical. What is, if the uptake observed in your uptake experiment is 90 % unspecific binding? Applying the standard characterization for radioligands would easily provide an answer to this question.
Further, also in point 3 the authors rise the following question:
“Honestly, authors do not understand what the Reviewer refers citing ref. 15 since in that publication the cell uptake was performed exactly in the same way described herein and no blocking study with curcumin was performed.”
In reference 15, the authors describe a similar characterization of a Gallium-curcurmin and reference is given to Antunes et al. (Doi.: 10.1007/s00259-006-0317-x); and Fani et al. (Doi.: 10.2967/jnumed.111.087999). Both, Antunes and Fani, describe the evaluation of their ligands including the determination of unspecific binding in their respective uptake studies. Further, in both references the logP of the respective ligands is investigated. It is to the best of the knowledge of the referee the accepted standard procedure for the preclinical in vitro evaluation of radioligands. The referee assumed that the authors were correctly applying the cited assays. Based on the provided reply by the authors, this was obviously not the case. However, this fact does not provide a reasonable argument that the applied determination of uptake was proper.
Further points:
In the second paragraph of the reply the author’s state: “High binding of the curcuminoid-like structures to plasma proteins is well established in the literature, therefore the authors deem further experiments un-necessary to confirm these results.”
It is reasonable qualified guess with respect to the literature. However, it is by no means the quantitative data required for the characterization of a novel pharmaceutical. Especially since the authors speculate on the contribution of plasma binding to the high blood pool activity this should have been quantified. However, since the compound is also metabolized, it is even unclear if the compound itself or some metabolite is binding to serum albumin (or the activity just accounts for a polar metabolite). In summary, nothing can solid be concluded from the observed blood pool activity.
Third paragraph of reply:
The referee agrees, that an investigation of a possible VDR mediated uptake is interesting for the characterization of the ligand. However, this can by no means replace the determination of unspecific uptake (see major point above).
The referee admits, that the uptake of 68GaCl3 is significantly different to the novel ligand 68Ga-DOTA-C21. However, this does not provide any information whether the observed uptake is specific or unspecific. The ligand could as well be bound to the surface in a non-specific manner or the uptake could be by passive diffusion over the membrane (see major point above).
Fourth paragraph of reply:
The referee agrees, that major deviations between the real pharmacokinetics and the in vitro based expectations commonly occur. However, before conducting an in vivo experiment the researcher has to be sure to gain reasonable knowledge. From the data presented in the manuscript it is not possible to conclude whether the uptake is specific or not. Eventually, it could also be a metabolite which is binding to the tumor or plasma protein bound radioligand which was taken up in the tumor via an unspecific EPR pathway. Since this cannot be clarified based on the provided data and, further, should have been known before conducting the animal experiment the referee remains the opinion, that the in vivo experiment was unethical.
The authors should at least consider raising the missing data now. If binding of the radioligand is unspecific, the animal data should not be published. If the uptake is specific the data can be published. However, the referee would even recommend to conceal the fact that this data after the animal experiment.
I know that the other referees obviously do not share my position on this point. However, to the best of my knowledge I am sure that my point of criticism on the specific binding is justified. Since this raises an ethical issue, I deeply regret that I did not change my opinion of this largely outstanding manuscript an still have to recommend rejection.
Author Response
The authors acknowledge the reviewer for the further comments and opinions. The point of the reviewer is well taken: further studies are necessary to understand specificity and mechanism of uptake. However, this is a first “chemistry” paper, largely focused on synthesis and characterization of a new tracer, therefore submitted for publication in a chemistry focused journal. In vitro data suggest specific uptake. In vivo imaging suggests uptake – which, as the reviewer must know, is not always the case with new PET radiotracers. We will study this tracer and other derivatives further in the years to come, and years will be needed given the unknown around curcumin and curcuminoids mode of action !
Regarding the reviewer’s last sentence i.e “ […] I did not change my opinion of this largely outstanding manuscript and still have to recommend rejection.”, the authors do not understand why the reviewer believes that a largely outstanding paper, focused on chemistry, should be rejected by a chemistry journal.